# Evolution of the Physical and Social Spaces of ‘Village Resettlement Communities’ from the Production of Space Perspective: A Case Study of Qunyi Community in Kunshan

**DOI:** 10.3390/ijerph16162980

**Published:** 2019-08-19

**Authors:** Lei Zhou, Liyang Xiong

**Affiliations:** 1School of Architecture and Urban Planning, Nanjing University, Nanjing 210093, China; 2School of Geographic and Biologic Information, Nanjing University of Posts and Telecommunications, Nanjing 210023, China; 3School of Geography, Nanjing Normal University, Nanjing 210023, China; 4Key Laboratory of Virtual Geographic Environment (Nanjing Normal University), Ministry of Education, Nanjing 210023, China; 5Jiangsu Center for Collaborative Innovation in Geographical Information Resource Development and Application, Nanjing 210023, China

**Keywords:** village resettlement community, production of space, power, capital, social class, Kunshan

## Abstract

Village resettlement communities (VRCs) are a special type of urban community that the government has promoted considerably during China’s rapid urbanization. This study uses the theory of the production of space as a basis to explore the processes and mechanisms of the physical and social space evolution of VRCs through a case study of Qunyi Community, one of the largest VRCs in Kunshan. Questionnaires and semi-structured interviews were employed in this study. Results indicate that the coupling relationship between local government power and diversified capital is the fundamental reason that promotes the production of macrophysical space. Moreover, the economic and social relationships among residents promote the reproduction of microsocial space. Landless farmers are the most important spatial producers in the microsocial space. The individual needs and cultural differences of immigrant workers also have significant effects on microspatial production. Furthermore, the production and reproduction of the physical and social spaces, respectively, of VRCs deduce the adjustment relationship among the urbanization processes of land, population, and individuals. Results also indicate that the urbanization of individuals appears to lag behind the previous two processes. This study can provide a theoretical basis for the spatial renovation and management optimization of VRCs, as well as the promotion of a new type of “people-centered” urbanization.

## 1. Introduction

Since the economic reform, China has experienced the largest and most rapid urbanization in human history [1,2]. As the largest developing country, China’s rapid urbanization has triggered the massive transformations of physical spaces and socioeconomic activities. For example, as plenty of agricultural land has been converted into urban use, rural farmers have been transformed to urban citizens, and rural villages have been changed into urban communities. However, the residents who have changed their respective status have yet to completely integrate into the urban lifestyle, and the urbanization of individuals lags behind that of land and population. Consequently, a wide range of social problems have inevitably emerged, such as unemployment, social inequality, and urban poverty. These challenges severely hinder the building of a “harmonious society” and sustainable development in China [3]. In 2014, the state government of China initiated a national campaign for a new type of “people-centered” urbanization [4]. The essence of this new urbanization is to synchronize social transformation with spatial urbanization.

The “village resettlement community” (VRC) is a special urban community that the Chinese government vigorously promotes in conjunction with rapid urbanization [5]. This special community should be a unique phenomenon of production of space during China’s urbanization process. From the perspective of international level, there are some other representative communities, such as urban villages [6] and slums [7]. However, the physical forms of these communities and their social organizations of residents are totally different from those of the VCRs because these communities are shaped by certain urbanization processes with different environmental backgrounds in different countries. At present, with the increasing number and expanding scale, VRCs have become a vital component of the urban space. Compared with previous rural communities, VRCs have undergone substantial changes in the physical community form and social organization of the residents. However, VRCs are centralized resettlements of landless farmers and often present a special city–village characteristic that is very different from other urban communities. In the context of the rapid urbanization in China, VRCs have become an important platform to evaluate the adjustment relationship among the urbanization of land, population, and individuals.

VRCs have drawn a great deal of attention from economists, social scientists, and urban planners, among others [5,8,9,10,11,12,13]. Economists focus on land compensation, social insurance of landless farmers, and the restructuring of collective assets [3,8,11]. Social scientists concentrate on the identity and community integration of landless farmers, and the social support network of VRCs [10]. Moreover, social scientists also focus on the operation and management issues of the community and its social problems [13]. Urban planners mainly focus on the governing advice and reform strategies of VRCs from the perspective of space [5,12].

In terms of research contents, the academia has generally provided comprehensive attention to the space, land, institution, and society of VRCs [13]. However, VRCs are exposed to many problems because of the complicated construction process and resident composition [11], thereby considerably limiting the development and transformation of these communities. Compared with the traditional rural village and urban community, VRCs show evident differences in terms of governing patterns, which remain under a trial-and-error process. Accordingly, the management of VRCs urgently needs a valid theory to interpret the logic of spatial production. The spatial construction and evolution of VRCs are processes that involve government power, multiple capital, village collective, and residents [8,9]. The theory of the production of space emphasizes the role of power and capital in shaping and producing physical space, and the reproduction of social space reshaped by social class [14]. The processes and mechanisms of the production of space in VRCs are unique because of the influence of power, capital, and diverse social classes in the rapid urbanization. However, minimal attention has been given to VRCs from the perspective of the production of space, which has left a research gap to be filled. It is easy to distinguish the social interaction behaviors of the participants and their impacts from the perspective of the production of space, thereby possibly contributing to the formulation of spatial management policies.

Kunshan City has been regarded as an advanced manufacturing base in the Shanghai-centered Yangtze River Delta in China, which is a representative city of land acquisition and VRC construction for the development zones [15,16,17]. Qunyi Community in Kunshan City is selected as the case study area of this research for its numerous landless farmers and migrant workers. Through a series of field observations, questionnaire surveys, and semi-structured interviews conducted in 2009–2016, we sought to study the evolution process and mechanism of the physical and social spaces of VRCs. The roles of power, capital, and social class were analyzed and their resultant impact on the community were evaluated. This research would have tremendous implications for the benign development of VRCs in China’s new type of “people-centered” urbanization.

## 2. Research Context and Theoretical Framework

### 2.1. Village Resettlement Community

In China, local governments have implemented a variety of projects to accelerate the urbanization process [18]. The VRC is a commonly used project by them. This specific project starts with the land requisition of a suburban village to meet the spatial expansion needs of the city [9]. Rural land acquisition in China is undeniably subjected to local governments [12]. Through land acquisition and administrative division realignment, local governments change farmers’ household registration status (hukou) from agricultural to non-agricultural, enroll them in the urban social security programs, and re-settle them to urban residential communities [9]. Meanwhile, the administrative body is transferred from a village committee to a neighborhood committee. This type of resettlement community, which comprises landless farmers, is called a VRC. It is important to point out that VRCs are neither rural settlements nor urban communities. They can be categorized as a transitional community type between rural settlement and urban community during China’s rapid urbanization. In a VRC project, the lands that farmers have lived on for generations are expropriated and the houses built on their own homestead are demolished. Once the land of a village is going to be expropriated, the local government immediately plans and constructs a cluster of high-rise apartments to resettle the farmers. Local government recompenses the farmers, and also provides relocation compensation and settlement allowance to the people whose houses are demolished [11].

The building of VRCs should be a systematic project that includes new community construction, resident identity conversion, and extension of public services [3]. Yet, it is often faced with many management obstacles, which are the most prominent problems in the current urbanization in China. In a VRC project, a few institutional barriers, such as the conversion of resident identity, are eliminated for urban development, thereby considerably accelerating the pace of urbanization. However, VRCs present unique characteristics, particularly given the high density of building volume, large-scale rental economy, abundance of unemployed residents, complex resident composition, and hidden dangers of public security [3,9,11].

### 2.2. Theory of the Production of Space

“The production of space” is a key concept of neo-Marxist urban school and Marxist geography [19]. Before the 1970s, space was supposed to be an abstract or physical concept that was unrelated to social concepts [20,21]. In 1974, Lefebvre creatively proposed the theory of “the production of space,” formulated the idea of “(social) space as a (social) production,” and believed that the relationship between space and society should be revalued [21]. This theory indicates that space is a backdrop and product during the entire process of social (space) production [22]. Since Lefebvre’s publication of The Production of Space, many scholars have continuously developed and applied this theory to the discussion of urban issues [23,24,25,26,27,28,29]. Several new frameworks have also been developed, such as the theory of “the urbanization of capital” [25] and the theory of “power space” [30,31], thereby making “power,” “capital,” and “social class” the key elements for explaining the production of space. Harvey presented that capital invested in a built environment is the major determinant of the production of urban space [20]. The uneven investment of capital produces uneven spaces, such as those in developing and developed areas [25]. Foucault proposed that space is a metaphor and symbol of power, and the shaping of space follows the logic of power [30,31]. For the social class, the view that space is a production of social life is the cornerstone of contemporary social and cultural geography [21,27,31,32]. Moreover, the idea that space can be shaped from the social meanings of people’s lives has drawn considerable attention [27]. For example, high- and low-income groups agglomerate in different living spaces during the process of urbanization. Thus, space can be used to separate social classes [28].

The production of space in urbanization has drawn considerable attention in terms of empirical research, thereby showing a “scale effect.” The research objectives and contents of the production of space in urbanization presents evident differences at various spatial scales. In the studies of the production of space at a micro-scale, scholars have focused on the production of social space in urbanization. Micro-scale space with certain characteristics of social structure, such as urban villages, rural–urban frontiers, affordable residential areas, “Taobao villages,” and university towns, have been taken as the research objects [14,33,34,35]. By applying a qualitative analysis method of sociology and behavioristics, these studies discuss the production of social space, which is promoted by collective actions driven by a social network relationship from a micro perspective. For the studies of the production of space at middle- or macro-scale, the production of physical space during the urbanization draws more attention, which focuses on the role of institution reform, power structure, and capital circulation [26,36,37]. For example, Shanghai, Paris, Baltimore, and Los Angeles were adopted to explain the process of production of space [22,38].

However, limited focus has been paid to the spatial production of VRCs compared with the aforementioned studies, which completely demonstrated that the theory of production of space effectively explains urban spatial phenomena and spatial processes. The construction and development processes of VRCs include the production of macro-scale physical space affected by power and capital and micro-scale social space shaped by residents. Thus, this theory would profoundly explicate the processes and mechanisms of the spatial production of VRCs.

### 2.3. Conceptual Framework

In this paper, we follow the theory of the production of space to explore the evolution of the physical and social spaces of VRCs (Figure 1). Power, capital, and social class have been regarded as the key elements of production of space. Specifically, power and capital affect the land through spatial construction (i.e., planning and site selection, demolition and resettlement, and investment and construction). These two factors have been regarded as the major factors that control the production of the macrophysical space of VRCs. In addition, the diverse social classes of multiple residents affect the community through spatial reshaping (i.e., lifestyle change, rise of informal economy, and social network reconstruction). The roles of diverse social classes have been regarded as the major factors that influence the production of the microsocial space of VRCs. The roles of power, capital, and class together form the conceptual framework for this study (Figure 1). With this framework, the VRC of Qunyi Community in Kunshan City is used for conducting the research on its evolution of physical and social spaces from the production of space perspective.

## 3. Study Area, Data, and Methodology

### 3.1. Study Area

The county-level city of Kunshan, which is within one hour’s drive to Shanghai, is located in Sunan Areas (part of the Yangtze River Delta and one of the most rapidly growing city regions in China) (Figure 2). Kunshan has a land area of 928 km^2^ and comprises 10 towns, 3 national-level development zones, and 2 provincial development zones (Figure 2). Before the 1980s, Kunshan was a typical agricultural county. As the richest county in China, Kunshan has undergone substantial economic development since the economic reform [39]. Kunshan maximized the benefit of its proximity to Shanghai and was the first to set up a development zone without fiscal subsidies from the central state, which was formally approved as a national development zone by the state council in 1992. The remarkable achievements of the development zone have enabled Kunshan to become one of the top 100 counties in China for years [15,39]. In 2014, Kunshan became the first county-level city with GDP of over 300 billion yuan.

Qunyi Community, which is located in the rural–urban fringe of Kunshan and the inner area of Kunshan economic and technological development zone (KETDZ), is the foreground of urbanization in Kunshan (Figure 2). Moreover, Qunyi Community is the largest VRC in Kunshan with 123 six-floor residential buildings and covers a total area of 11.7 km^2^. This community comprises three residential areas, namely, North, East, and West Zhong-hua-yuan Villages. In May 2002, to accelerate the urbanization of KETDZ, the government completed the transformation of the urbanization institution system, repealed the Qunyi Village Committee, and established the Qunyi Community Committee. Correspondingly, farmers were transferred to urban resident status, while collective-owned rural lands were converted into state-owned lands. As a land requisition compensation, landless farmers were relocated to a modern urban community called Qunyi Community from their village houses. By the end of 2002, 5100 farmers from 1228 households were resettled to the newly built Qunyi Community and granted urban citizenship. The new residents have remolded the space of the community on their own, thereby providing it with distinguished characteristics different from those of urban communities. The spatial production of this VRC was completed approximately 10 years after the residents began to move in. Thus, Qunyi Community is a good case to study the production of space of VRCs.

### 3.2. Data Sources and Methodology

Data were obtained from field observations and interviews with the urban planning administration, management committee of KETDZ, and Qunyi Community Committee with three different time periods (i.e., September 2009, June 2013, and March 2016). The interviewees were officials, community administrators, and residents. To investigate the production process of physical space of the VCR, the interview contents include the planning and policy measures and construction processes of Qunyi Community. In addition, to explore the production process of social space of the VCR, a questionnaire survey was also conducted among the residents in March 2016 (Appendix A). The interviewees were classified into four types: Landless farmers, migrant workers, local New Kunshan citizens (i.e., farmers in the surrounding villages who purchased second-hand housing in Qunyi), and alien New Kunshan citizens (i.e., migrant workers who purchased second-hand housing in Qunyi). The detailed information collected through the questionnaire includes the residents’ personal attributes, personal and family members’ occupation, workplace, and income (Table 1). The housing or renting conditions are also included in the questionnaire (i.e., length of residency, housing or renting area, satisfaction with the environment, number of resettlement apartments, relationship with the tenant, attitude toward the tenant, and rent) to analyze the living and consumption space in the VCR (Table 1). We also surveyed the social interaction and daily activities of the residents to explore the communication space in the VCR. A total of 232 out of the 250 questionnaires were valid (effective rate of 92.8%), thereby meeting the sample size requirements.

## 4. Production of Physical Space

### 4.1. Powerful Propulsion of the Local Government

The majority of the cities in China have prioritized economic development. The municipal government of Kunshan has also utilized KETDZ as an important instrument to attract foreign investments and enhance Kunshan’s economy. With its monopolistic and compulsory administrative power, the Kunshan government generally controls the allocation of financial and land resources, which serves the development of KETDZ. In 2000, Kunshan set up the first export processing zone (EPZ) in China, thereby further expanding the spatial scale of KETDZ. Meanwhile, the spatial expansion direction of *KETDZ* shifted from the east of the old town to the south (Figure 3). Suburban areas, such as Qunyi, Chezhan, and Xiongzhuang Villages, which were originally located near the EPZ, became the major locations of new buildings and constructions of the EPZ (Figure 3).

In China, all urban lands are owned by the state, whereas rural lands are owned collectively and cannot be used for the construction of development zone. The only legitimate way of utilizing collective-owned rural lands is land requisition initiated by local governments [9]. Farmers’ rights to collective-owned lands, including farmland and housing plots (zhaijidi), are forcibly expropriated by the government. Thus, the Kunshan government adjusted its administrative boundary. The entire territory of the three villages was transformed to state-owned urban land for the construction of EPZ (Figure 3). Moreover, the Kunshan government launched the VRC project. To accelerate the implementation of this project, various levels of local governments were combined and constituted a project-oriented regime (i.e., from high to low: Kunshan County government; administration committee of KETDZ; township governments, such as Qunyi, Chezhan, and Xiongzhuang Villages; and Zhong-hua-yuan Sub-district Office).

Once the entire village is demolished (including houses and croplands), local states should construct urban apartment communities to accommodate the displaced landless farmers. Every household can receive several apartments with various sizes as compensation, depending on the household’s size, single-child status, and other standards. Prior to the demolition, a new community, namely, Qunyi Community, was built in the south of KETDZ, which is near the three original villages, to accommodate the relocated villagers (Figure 3). The construction of Qunyi Community commenced in June 1999 when the Kunshan Planning Bureau started to seek a site for the resettlement of the farmers from the three villages. In April 2000, the demolition of the three villages started immediately after the construction of Qunyi Community. The formidable tasks, which include demolishing the villages and relocating the landless farmers, were assigned to Zhong-hua-yuan Sub-district Office. Numerous “strategies” were used to guarantee the rapid demolition and relocation, including such incentives as compensation reward and job placement (by interviewee NO.1, government official, 2009). Indeed, relocating the 5500 displaced farmers in such a short period of time remained a considerable “achievement.”

### 4.2. Joint Operation of Capital and Power

The construction of VRCs is state sponsored and top-down. However, this process also involves the spatialization of capital, which is highly dependent on policies and funds, and should be operated jointly through government power and diversified capital. The local government plays an important role in the spatialization of capital by monopolizing land resources and land development right and regulating the development direction of land. On the one hand, the Kunshan government accumulated industrial surplus capital through investing in local infrastructure, improving the investment environment, and attracting additional foreign capital. On the other hand, by utilizing the industrial surplus capital, the government has expropriated land and conducted the VRC project and resettled farmers from the traditional village community to urban community, thereby realizing the second loop of the capital.

The implementation of the VRC project needs the cooperation and collaboration of numerous stakeholders, such as local states, local banks, and local communities, to create an efficient coalition. The Kunshan Urban Construction Investment Development Group Co. Ltd. (KUCIDGCL), set up by the state-owned Assets Supervision and Administration Office of the Kunshan state, was directly responsible for the VRC project. KUCIDGCL is a wholly state-owned enterprise. Government authorization states that this company was responsible for financing the construction of Qunyi Community and the entire process of land expropriation, demolition, resettlement, and project financing and loan repayment. By relying on the resources allocated by the Kunshan government, KUCIDGCL absorbed capital through various financing methods, such as land transfer, land mortgage, and enterprise mutual guarantee, to provide strong financial support for the construction of Qunyi Community. A total of approximately RMB 3.8 billion yuan was invested to the VRC project, in which the Kunshan government was the major and direct investor (by interviewee NO. 2, government official, 2009). The construction of Qunyi Community cost approximately RMB 1.1 billion yuan. Moreover, basic infrastructure, such as roads and water supply and drainage system, cost approximately RMB 0.9 billion yuan. Evidently, local banks were critical in the collation of investments. Other investments were used for the demolition of villages and compensation purposes. Every relocated household received a compensation between RMB 200,000 and 260,000 yuan. The construction of Quyi Community guaranteed the relocation of farmers, thereby becoming a major stimulation for the success of the relocation. Although parts of the main construction lasted until November 2001, the majority of the structures were completed before the proposed deadline of 1 May 2001. As a megaproject, Qunyi Community involved the simultaneous construction of 123 buildings. However, the entire project needed merely 30 months. The strong leadership of the local government relatively compensated for the deficiency in capital, thereby facilitating the efficient construction and formal management of the VRC.

## 5. Reproduction of Social Space

### 5.1. The Changing Composition of Community Residents

The evolution of residents in Qunyi Community involved two stages: Agglomeration of migrant workers from 2002 to 2010, and out-migration of landless farmers since 2010 (Figure 4).

Initially, the residents of Qunyi Community were all local farmers. As a VRC, Qunyi Community was connected by a traditional social network based on kinship and geography. As compensation for farmhouse requisition, each affected household was compensated with two to four apartments, approximately 85 m^2^ of each in Qunyi Community. One or two of the apartments is/are used for living by the farmers themselves; the other apartments are allowed to be rented or even sold. The foreign-invested enterprises in KETDZ have attracted massive labor migration. Qunyi Community became extremely popular with migrant workers because of the low rent and cost of living and its proximity to KETDZ. Migrant workers are young, mainly aged under 35 years (Table 1), and are generally from northern Jiangsu and Anhui provinces. In addition, the farmers of the surrounding villages of Kunshan City opted to buy second-hand apartments in Qunyi Community, thereby becoming the local New Kunshan citizens. A few migrant workers who had already decided to settle down in Kunshan also purchased second-hand housing in Qunyi Community, thereby making them alien New Kunshan citizens. Moreover, the displaced farmers are gradually differentiated into different social classes because of their varying occupations and income. A few of the high-income households sold their apartments in Qunyi Community. Thereafter, they bought commercial apartments, moved into high-grade established urban communities, and became “urban residents” in the real sense of the term. The first-generation middle-income landless farmers bought commercial apartments for their children near Qunyi Community. On their part, they continue to live in Qunyi Community. Therefore, the long-lasting spatial proximity among the fellow landless farmers is gradually disintegrating. As the “outsiders” increase, Qunyi Community experiences a huge change of becoming a mixed residential community for landless farmers, migrant workers, and the local and alien New Kunshan citizens. The proportion of local residents shrinks, whereas that of migrants increases substantially. According to community managers, approximately 20,000 residents lived in Qunyi Community in 2016; however, the number of non-agricultural Kunshan locals is merely 5800, accounting for only 29% of the residents of Qunyi. The migrant workers in Qunyi Community, who were temporary dwellers and highly mobile, were over twice as many as those of the local residents.

### 5.2. Changes in the Social Relationships among Residents

The massive influx of migrant workers has brought about huge socioeconomic changes for Qunyi Community. The original close social ties based on kinship and geography have been continuously eroded and gradually replaced by a clear and common economic tie. Qunyi Community gradually presented a hybrid social form.

#### 5.2.1. Close Economic Relationship

In China, each rural household is allocated with a parcel of farmland by a contract, which is taken as the primary means of livelihood [40]. The VRC project separates the farmers with their agricultural land. Once the means of livelihood has been deprived, the landless farmers have to find other means to earn a living. For these farmers, the VRC project should be a process that involves “being urbanized.” They can hardly engage in non-agricultural work and integrate into urban life. The assistance offered by the Qunyi Community Committee led to a few displaced farmers being hired by foreign-invested enterprises as service workers. These working positions mainly focus on weeding, cleaning, and guarding (by interviewee NO. 3, community administrator, 2013). Still, nearly half of the landless farmers are unemployed. Thus, they have to look for other means of living. The surplus resettlement apartments received as a form of compensation have become an important source of income for the landless farmers and can be rented for a considerable amount (by interviewee NO. 3, community administrator, 2013). The survey found that all the households of landless farmers run a rental business. The quantity of tenement apartments per household varies from one to three. In addition, the average monthly rent per apartment is approximately RMB 2300 yuan (by interviewee NO. 4, accountant in the community, 2016). The rent paid by migrant workers greatly raised landless farmers’ income.

However, service sectors around Qunyi Community that are located on the edge of the main urban area of Kunshan lag behind the residential development. Except for several supermarkets around the community, only a few service facilities have been available in recent years. However, the migrant workers in Qunyi Community have huge consumption and leisure demands, thereby providing a favorable environment for landless farmers to spontaneously run informal business. To earn additional income, the landless farmers are no longer confined to the “rental economy”. They also opened stores, restaurants, hotels, studios, and Internet cafés to fill in the service vacuum in this community. Therefore, this community has turned into a primary place for the provision of middle- or low-end services for migrant workers. Migrant workers are optimistic to continually have a low cost of living space, whereas the landless farmers actively pursue the maximization of rents and earnings. The residents in Qunyi Community show a good symbiotic economic relationship under the influence of the mutually beneficial economic ties.

#### 5.2.2. Well-Defined Social Relationship

In the questionnaire survey, the question “Who do you usually associate with?” was asked to investigate the social network of the residents. The finding shows that the social network of the landless farmers remains based on kinship and geography (Table 2). Migrant workers often associate with their co-workers, friends, and fellow villagers. Their social network based on work and geography formed spontaneously. On the one hand, affected by the homeland feelings, the migrant workers who have taken root in this community constantly expand their social network to seek more fellow villagers. On the other hand, for those migrants who hope to work in Kunshan, they contact and associate with their fellow villagers to help them find a job. Apart from the mutual assistance on career, migrant workers also help one another in their daily lives (e.g., introducing residences), thereby further strengthening their social network. The majority of the local New Kunshan citizens came from the surrounding villages of Kunshan. These residents communicate daily with their relatives and friends, thereby making their social network also based on kinship and geography. By contrast, the alien New Kunshan citizens, who have already bought apartments and settled in Qunyi Community, generally associate with colleagues, friends, and peers. Thus, their social networks are based on work and geography.

The question “Your views on the other three types of residents” was asked to study the social relationships of the residents. For the views of landless farmers on migrant workers, 45.5% of the interviewees said that they had good relationships with migrant workers, while 13.6% believed that migrant workers play an important role in the community economy. Moreover, 27.3% of the interviewees were optimistic that they can rent permanently. Less than 10% of the interviewees said that no difference exists between migrant workers and locals, while 63% of the interviewees regarded migrant workers as a special group. However, for the migrant workers’ view on landless farmers, over 50% of the interviewees said that their relationship with the landlords was general, while less than 1% thought that they get along with the locals harmoniously. In addition, 80% of the interviewees said that they are optimistic that they can transfer to a better apartment. Landless farmers were more optimistic about their relationship with migrant workers because of the “rent economy,” whereas the latter did not give an equal response. Except for tenancy relationship, numerous contradictions exist between landless farmers and migrant workers in Qunyi Community. With a feeling of discrimination, migrant workers failed to communicate proactively with landless farmers and found that integrating into the community culture is hard.

For the local New Kunshan citizens’ views on other residents, 45.5% of the interviewees said that they have a good relationship with the landless farmers. Over 60% of the interviewees said that their relationship with the migrant workers is not good, while 70% of the interviewees said that no difference exists between the landless farmers and them. The local New Kunshan citizens and landless farmers have a good relationship because their culture and dialect are nearly the same. However, the in-depth interview indicates that only 10% of the interviewees frequently communicate with the alien New Kunshan citizens or migrant workers. No economic relationship was found between the local New Kunshan citizens and migrant workers. Thus, the communication between the two sides is minimal and the social relations are considerably distant. The locals deem that the majority of the migrant workers are uneducated, uncivilized, ignorant, and unsanitary. The migrant workers are blamed for the increased crime and degradation of sanitation in the community. Consequently, such hostile, suspicious, and discriminatory attitudes toward migrant workers have led the locals to avoid day-to-day contact with them. For the alien New Kunshan citizens’ opinion on the other residents, 70% of the interviewees said that their relationship with the landless farmers and local New Kunshan citizens was general, while 81% of the interviewees said that they had good relationships with migrant workers. In addition, 55% of the interviewees expressed a desire to move to a better community. The alien New Kunshan citizens are minimally integrated into Qunyi Community. They only consider this community as a transitional residence in Kunshan. However, the alien New Kunshan citizens and migrant workers have a close relationship because of their same identity as migrants. In general, the various residents in Qunyi Community have formed a distinct social network and relationship. The frequency and depth of communication between the natives and the migration or the migrant workers in different regions are relatively low.

### 5.3. Reproduced Social Space

#### 5.3.1. Living Space

With a high unit-to-plot ratio, the residential buildings in Qunyi Community show a barracks-style spatial distribution pattern. Apart from the garage and storage rooms on the first floors, the remainder of the residential buildings are used as living spaces. Landless farmers have a large living area and good living environment. However, for the rental apartments, the landless farmers divide the original rooms to increase the number of units available for rent and to procure additional rents (Figure 5).

In each rental apartment, many types of living space are available, which cost RMB 400 yuan to 800 yuan per room monthly. For example, 10 m^2^ to 20 m^2^ rooms (can be divided into a room with windows and no windows) are rented by unmarried migrant workers, while suites above 20 m^2^ are rented by migrant families. Therefore, several persons and even several families share an apartment. For example, at the No. 32 building in East Zhong-hua-yuan Village, only 12 among the 36 apartments were used by landless farmers for self-living, while 3 apartments were bought by migrants. In addition, 21 sets are used for rental, in which 4 sets are used by landlords and tenants living together, 14 sets are rented completely, and the remaining 3 sets are remolded into simple hotels. Thus, the population structure of Qunyi Community is complex and the residential density increases. The architecture landscape of the living space appears to be increasingly cluttered. Hygienic disorder occurs with certain security issues and hidden dangers, which have a negative effect on the comfort and safety of the living space.

#### 5.3.2. Consumption Space

The supply and demand relationship of landless farmers and migrant workers promoted the conversion of original garage and storage rooms on the first floors of the residential buildings into informal commercial stores (Figure 6). Landless farmers created all types of consumption spaces for migrant workers. The differentiations of the cultural and consumption demands of migrant workers gave rise to a variety of consumption space with local characteristics, such as all types of local cuisine restaurants, labor and rent intermediaries, and Internet cafés. Such differentiation has undoubtedly caused the heterogeneity of the production of space in Qunyi Community (Figure 7).

Interweaving living and consumption spaces are important spatial characteristics of this community. The pursuit of economic benefits of landless farmers and the consumption demands of migrant workers have jointly promoted the production of “micro space” in the community. The landless farmers are the most important producers of the “micro space.” Landless farmers have changed the function of the space in the community, which has caused chaos in the first-floor spaces of the residential buildings. Accordingly, the width of the streets in front of the apartment buildings narrowed in Qunyi Community, and the sanitary environment became messy with hidden dangers.

#### 5.3.3. Communication Space

Before the VRC project, the communication space of landless farmers was mainly concentrated on traditional courtyards, village stores, and fields. The barracks-type spatial distribution pattern of the residential buildings in Qunyi Community has restricted the interaction between the residents. Heavily influenced by the traditional rural lifestyle, the psychological status and living habits of landless farmers could not fit in with a completely different urban habitat. This has led the landless farmers to voluntarily remold the urban community to spaces with functions that they were familiar with in their original village houses (by interviewee NO. 5, community administrator, 2013). Consequently, grocery, mahjong, and chess rooms converted from the first-floor garages became public places of social interaction for the landless farmers, which not only satisfy the daily consumption need of the residents, but also provide the landless farmer with a communication space (Figure 8). Moreover, the entryway of every building gradually turned into another gathering spot, where the landless farmers can talk or play cards together (Figure 8). Although the consumption space has created an atmosphere of social interaction for the residents, different consumption spaces were targeted for different consumers, which also became an obstacle for the interaction between landless farmers and migrant workers. For example, employment agency, game halls, billiard halls, and Internet cafés, which served the migrant workers in Qunyi Community, became an important communication space for them. However, the landless farmers seldom visit these places. The differentiation of communication space further aggravates the segregation among the residents. An invisible wall separating the locals and migrants existed in the VRC. This sociospatial segregation would not hinder landless farmers in earning rent from migrant workers. Instead, it protects landless farmers’ way of living from the interference of large-scale immigration.

## 6. Discussion

The production of space of VRCs is not only a construction process of the external landscape environment, but also a reproduction process of the social space. The case study of Qunyi Community indicates that the remolding of living space and establishment of new consumption spaces are signs of the establishment of new economic relationships and social structures. During this process, not only space, but also a social boundary has been produced. At present, landless farmers and migrant workers live in urban community, engage in non-agricultural occupation, and attempt to follow an urban lifestyle. However, they still cannot integrate to the urban life that is reflected in the evident sociospatial boundaries among different resident categories. For the migrant workers, they lack a sense of belonging and security. The urbanization of lifestyle of individuals is the main target of the new type of “people-centered” urbanization in China [4]. The production of physical space of VRCs shows that the urbanization of land and population in China could be planned and manipulated by the government. However, the urbanization of individuals, which is represented by the production of social space, is difficult and challenging. In China, within the process of massive urban expansion by the means of land acquisition, the urbanization of land and population is relatively easy. However, the urbanization of individuals often lags behind the previous two. The social relationship and spatial boundary of the residents in VRCs are actually typical microcosms of the urbanization of individuals lagging behind the urbanization of land and population. For landless farmers and migrant workers, how to communicate with one another and integrate into the urban life and how to create an open and inclusive community with distinct local sense and cultural atmosphere are areas worthy of further study in China’s new type of “people-centered” urbanization.

## 7. Conclusions

As a special urban community, VRCs are extensively promoted by the Chinese government with a strong political power in the rapid urbanization. With the agricultural lands rapidly transformed to urban use and the household registration status converted from agricultural to non-agricultural, the extensive spatial transformation of VRC also triggered the restructuring of social relations of the residents, which in turn reshaped the space of the VRC. This study uses the theory of the production of space as a framework to consider Qunyi Community in Kunshan as a case study and analyze the production and reproduction of the physical and social spaces, respectively, of VRCs in China. We attempt to investigate the spatial production process of VRCs from the perspective of power, capital, and social class. Accordingly, we would deepen our understanding of the matching relationship among the urbanization of land, population, and individuals in China.

The spatial production of VRCs is a process of “top-down” production of macrophysical space dominated by government power and multiple capital and a “bottom-up” production of microsocial space shaped by multivariate residents. In the rapid urbanization of China, the VRC project is a de facto state-led project that is completely controlled by the local government. Furthermore, the local government determines the investment of capital. A solid alliance between state and market is established, thereby jointly promoting the production of the physical space of VRCs. In this process, landless farmers, faced with a strong government and developers, are passive during demolition, employment settlement, and housing compensation.

Given the massive influx of migrants and the changes in the occupation of landless farmers, a new socioeconomic relationship is gradually formed in a new physical space. The complex and diverse new social relationships among the residents have profound impact on social space, which is the most fundamental cause of the production of the microsocial space of VRCs. Moreover, VRCs have become heterogeneous living communities that are dominated by a migrant population and maintained by mutually beneficial economic ties. The diversified demands of migrant workers promote the microspatial production in VRCs. However, landless farmers are the main producers of the microsocial space. They run informal businesses that relied more on the spatial resources provided by the VRC. Their production of the social space is reflected in the reconstruction of the original living space and the construction of informal commercial space. Consequently, the microcosmic landscape of VRCs has been reshaped, thereby making it a depression among the urban communities.

## Figures and Tables

**Figure 1 ijerph-16-02980-f001:**
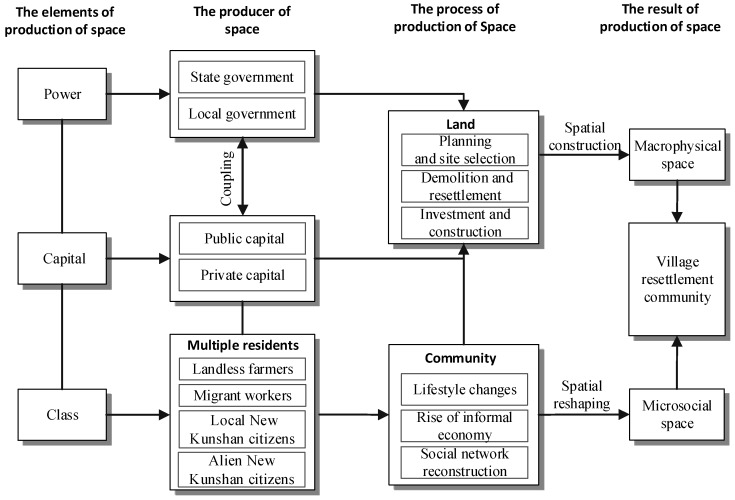
Conceptual framework.

**Figure 2 ijerph-16-02980-f002:**
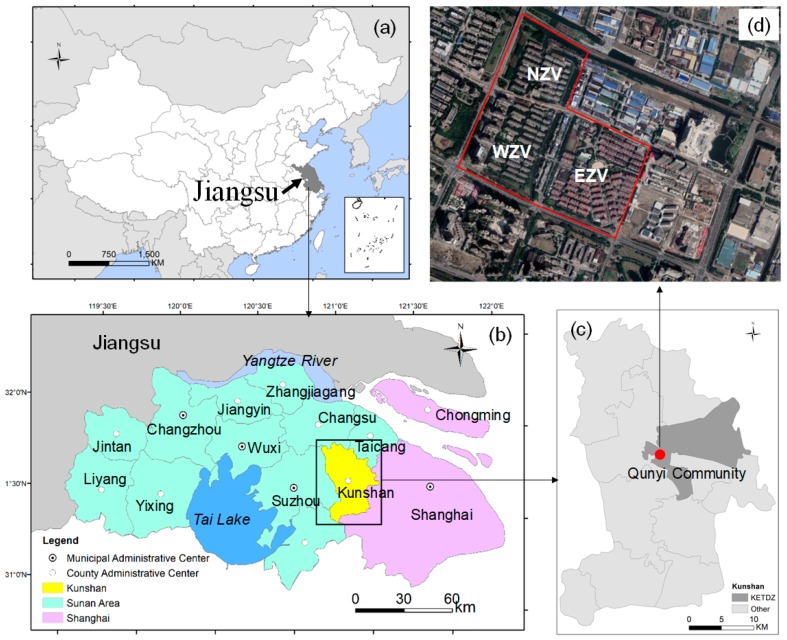
Study area: (**a**): Jiangsu province in China, (**b**): Kunshan in Sunan area, (**c**): Qunyi community in Kunshan, (**d**): Image of Qunyi community.

**Figure 3 ijerph-16-02980-f003:**
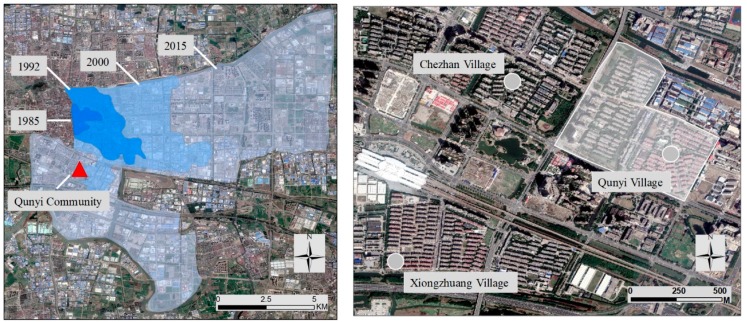
Spatial expansion of Kunshan economic and technological development zone (KETDZ) and the original location of Qunyi, Chezhan, and Xiongzhuang Villages.

**Figure 4 ijerph-16-02980-f004:**
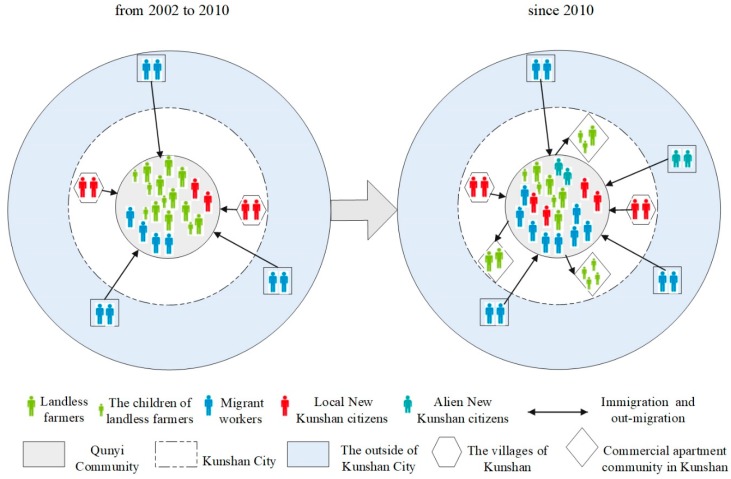
Evolution of the resident composition in Qunyi Community.

**Figure 5 ijerph-16-02980-f005:**
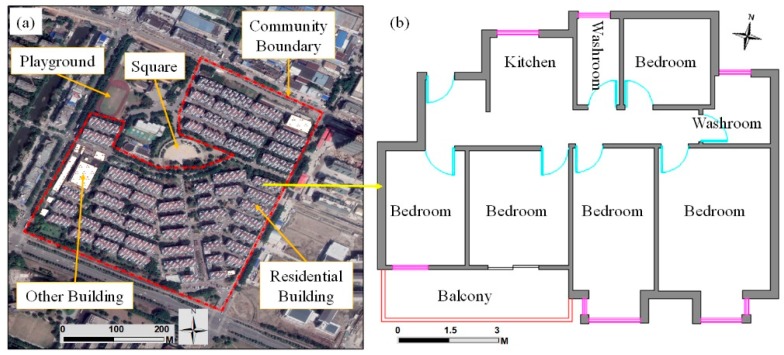
Living space of Qunyi Community.

**Figure 6 ijerph-16-02980-f006:**
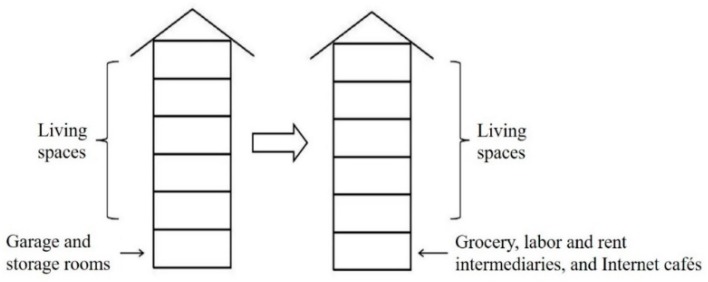
Renovation of the residential building space of Qunyi Community.

**Figure 7 ijerph-16-02980-f007:**
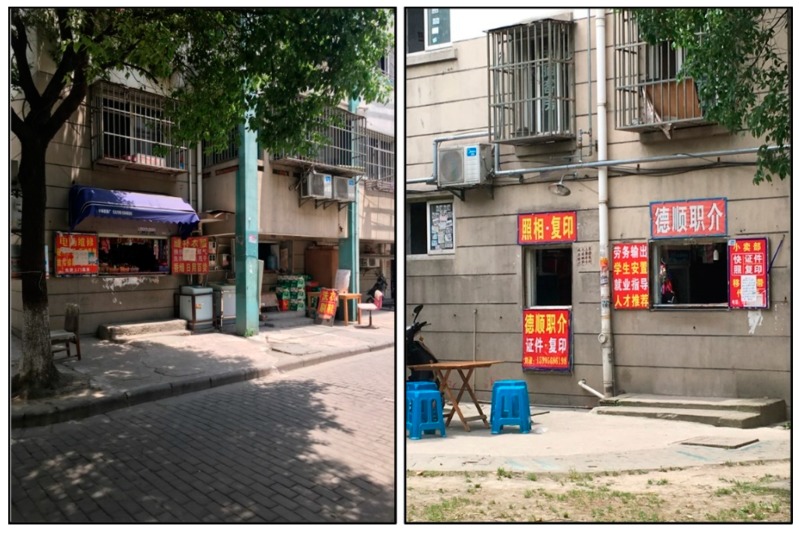
Consumption space of Qunyi Community.

**Figure 8 ijerph-16-02980-f008:**
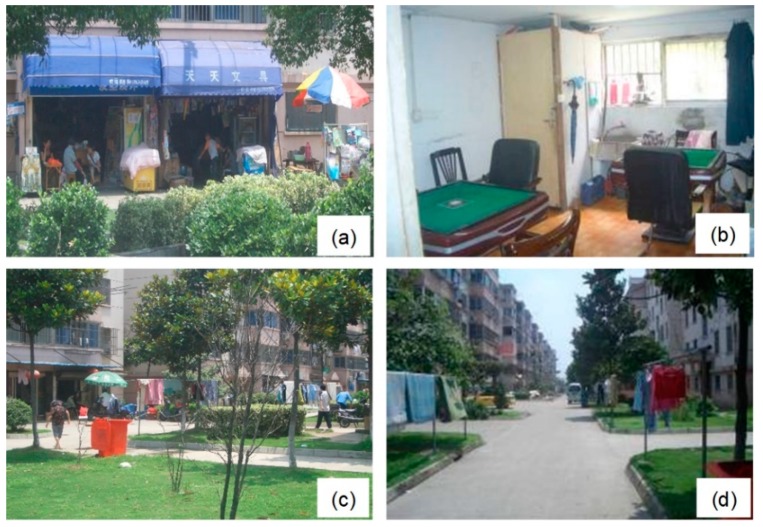
Communication space of Qunyi Community. (**a**) Grocery, (**b**) mahjong and chess room, (**c**) entryway of each building, (**d**) road in front of the apartment building.

**Table 1 ijerph-16-02980-t001:** Individual attributes of the surveyed sample.

Residents Type		Landless Farmers	Migrant Workers	Local New Kunshan Citizens	Alien New Kunshan Citizens
sample size		65	73	48	46
average age (years old)		55.3	31.2	42.1	39.8
length of residency (years)		14	3.2	7.9	8.4
marriage (%)	married	100	20.5	100	91.3
	unmarried	0	79.5	0	8.7
education level (%)	primary school and below	20.0	0	0	0
	junior high school	76.9	30.1	16.7	13.0
	senior high school /vocational high school	3.1	64.4	75.0	73.9
	junior college/ undergraduate and above	0.0	5.5	8.3	13.0
occupation (%)	government employee /public servant	4.6	0	4.2	2.1
	enterprise worker	70.8	95.9	72.9	54.4
	manager/ technical staff	0	4.1	10.4	21.7
	landlord	92.3	0	0	4.3
	businessman	20	0	12.5	17.5
	community manager	4.6	0	0	0
registration status (%)	local non-agricultural residence	100	0	52.1	71.7
	local agricultural residence	0	0	47.9	0
	nonlocal registered permanent residence	0	100	0	28.3
housing or renting area (%)	below 90 m^2^	30.8	0	16.7	89.1
	above 90m^2^	69.2	0	83.3	10.9
	below 10	0	63.1	0	0
	10 m^2^–20 m^2^	0	23.3	0	0
	above 20 m^2^	0	13.6	0	0

**Table 2 ijerph-16-02980-t002:** Characteristics of interviewees’ social networks.

Residents		Landless Farmers	Migrant Workers	Local New Kunshan Citizens	Alien New Kunshan Citizens
sample size		65	73	48	46
associate with (%)	relatives	100.0	16.4	100.0	17.4
	neighbors	35.4	12.3	8.3	2.2
	friends	66.2	86.3	77.1	65.2
	co-worker	18.5	91.8	27.1	41.3
	fellow villagers	58.5	98.6	18.8	19.6
	others	9.2	2.7	6.3	4.3

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
