# Peer review of "Evolution of the Physical and Social Spaces of ‘Village Resettlement Communities’ from the Production of Space Perspective: A Case Study of Qunyi Community in Kunshan"

_ijerph, 2019, doi:10.3390/ijerph16162980_

Round 1
Reviewer 1 Report
The aims are clearly described in the Introduction: "Through a series of field observations, questionnaire surveys, and semi-structured interviews conducted in 2009–2016, we sought to study the evolution process and mechanism of the physical and social spaces of VRCs.'
These aims are considered to be relevant, although the fitting into the aims&scope of the "International Journal of Environmental Research and Public Health" seems excessively forced and tangential, as health issues are practically non-existent in the text in its current version.
It is considered that the revised and improved paper could fit better in other MDPI journals, such as 'Urban Science', among others.
The following comments and suggestions are specifically made:
(1) The repercussions of the quality of life in the VRC should be deepened in a very notable way compared to the existing ones, and in the measure of the possible conditions of Public Health in the same ones.
(2) The concept of 'village resettlement communities' should be contextualised at the international level, providing similar examples in other territorial spheres or evidencing their differentiated identity.
(3) A detailed description should be given of the methodology used to obtain the information used in the text, description of the questionnaires, its analysis methodology, etc.
(4) It is recommended to deepen in the statistical analyses used for the evaluation of the surveys, the use of absolute frequencies is too simplistic for the problem presented.
(5) In the discussion, a critical evaluation of the authors of the urbanizing process as a whole is missed, specifically the process of urbanization of land suitable for agriculture, and consequently the expulsion and annulment of fundamental means of production. The criteria for selecting urban sites should be discussed. The criteria for selecting urban sites should be discussed. In good spatial planning, the conservation of fertile cultivated land is usually key, situating the new urbanized areas in usually less fertile land and with other qualities.
(6) Line 60 and 61: Review the term 'socialist'. The term 'social scientists' is probably more appropriate.
(7) Figure 3 right and figure 5 need graphic scale and North.
Author Response
The aims are clearly described in the Introduction: "Through a series of field observations, questionnaire surveys, and semi-structured interviews conducted in 2009–2016, we sought to study the evolution process and mechanism of the physical and social spaces of VRCs.' These aims are considered to be relevant, although the fitting into the aims & scope of the "International Journal of Environmental Research and Public Health" seems excessively forced and tangential, as health issues are practically non-existent in the text in its current version. It is considered that the revised and improved paper could fit better in other MDPI journals, such as 'Urban Science', among others.
Response: We do appreciate the reviewer’s comment. Actually, this paper has been submitted to a Special Issue of IJERPH, i.e., "Interplay Between the Environment and Health Issues in Cities of Developing Countries". The keywords of the Special Issue are urbanization, environmental health, One Health approach, basic urban services, environmental living conditions, and less developed countries. However, we believe the VRC in this paper should be a special urban space during the urbanization process in China. And the evolution process and mechanism of physical and social spaces of VRCs are quite related to the environmental living conditions of different types of residents. In addition, China is the largest developing country. Thus, we believe our paper is suitable for publishing in this Special Issue of IJERPH.
The following comments and suggestions are specifically made:
(1) The repercussions of the quality of life in the VRC should be deepened in a very notable way compared to the existing ones, and in the measure of the possible conditions of Public Health in the same ones.
Response: Actually, in the entire section 5 (i.e., Reproduction of social space), we were trying to discuss three repercussions of the quality of life in the VRC. These repercussions are the changing composition of community residents, the changes in the social relationships among residents, and the reproduced social space. For the first one we found that the evolution of residents in Qunyi Community involved two stages: agglomeration of migrant workers from 2002 to 2010, and out-migration of landless farmers since 2010. The comparison analysis between these two time periods help to understand the changing composition of community residents. For the second one, the analysis of close economic relationship and well-defined social relationship help to understand the huge socioeconomic changes for Qunyi Community due to the massive influx of migrant workers. With this understanding, the original close social ties based on kinship and geography have been continuously eroded and gradually replaced by a clear and common economic tie. Qunyi Community gradually presented a hybrid social form. For the third one, the living space, the consumption space, and the communication space among residents with different quality of life have been discussed to achieve a better understanding of the reproduction of social space. In addition, we personally believe these discussions are quite related to the Public Health in the VRC of Qunyi Community. Hope these clarifications can help to have a better understanding of this paper.
(2) The concept of 'village resettlement communities' should be contextualized at the international level, providing similar examples in other territorial spheres or evidencing their differentiated identity.
Response: Thanks for your kind suggestion. Yes, it is true that the concept of 'village resettlement community (VRC)' should be contextualized at the international level. Actually, the VRC is a special urban community that the Chinese government vigorously promotes in conjunction with rapid urbanization. This special community should be a unique phenomenon of production of space during China’s urbanization process. From the perspective of international level, there are some other representative communities, such as urban village, and slum. However, the physical forms of these communities and their social organizations of residents are totally different from those of the VCRs. Because these communities are shaped by certain urbanization processes with different environmental backgrounds in different countries. These details have been added in the revised manuscript (See lines 49-55)
(3) A detailed description should be given of the methodology used to obtain the information used in the text, description of the questionnaires, its analysis methodology, etc.
Response: First, about the questionnaire, this comment has also been pointed out by another reviewer. Thus, in the revised manuscript, the questionnaire has been provided as appendix of this paper as another reviewer suggested. In addition, we raised a new section of 2.3 which tries to describe the methodology of the framework. That is, in this paper, we follow the theory of the production of space to explore the evolution of the physical and social spaces of VRCs (Figure 1). Power, capital and social class have been regarded as the key elements of production of space. Specifically, power and capital affect the land through spatial construction (i.e., planning and site selection, demolition and resettlement, and investment and construction). These two factors have been regarded as the major factors which control the production of the macrophysical space of VRCs. In addition, the diverse social classes of multiple residents affect the community through spatial reshaping (i.e., lifestyle change, rise of informal economy, and social network reconstruction). The roles of diverse social classes have been regarded as the major factors which influence the production of the microsocial space of VRCs. The roles of power, capital and class together form the conceptual framework for this study (Figure 1). With this framework, the VRC of Qunyi Community in Kunshan City is used for conducting the research on its evolution of physical and social spaces from the production of space perspective (See lines 160-172). Finally, other information of the methodology can be find in the section 3.2 of data sources and methodology. In this section, the details of how we obtain the information have been provided (See lines 208-225). Hope these revision and clarification can help to have a better understanding of this paper.
(4) It is recommended to deepen in the statistical analyses used for the evaluation of the surveys, the use of absolute frequencies is too simplistic for the problem presented.
Response: Yes, it is true that the only use of absolute frequencies seems simplistic for the problem presented. However, apart from the absolute frequencies analysis, actually, we have conducted some necessary deepen analysis of the surveys. For instance, in section 5.1, we have discussed ‘the changing composition of community residents’ according to our surveys (Table 1). The detailed finding of the survey in this section can be found in lines 313-331. In addition, in section 5.2.2, we have also discussed ‘the well-defined social relationship’ according to our surveyed analysis of characteristics of interviewees’ social networks (Table 2). The detailed finding of this survey analysis in this section can be also found in lines 369-421. Hope this clarification of deepen analysis of surveyed data can help to have a better understanding of this paper.
(5) In the discussion, a critical evaluation of the authors of the urbanizing process as a whole is missed, specifically the process of urbanization of land suitable for agriculture, and consequently the expulsion and annulment of fundamental means of production. The criteria for selecting urban sites should be discussed. In good spatial planning, the conservation of fertile cultivated land is usually key, situating the new urbanized areas in usually less fertile land and with other qualities.
Response: Actually, as the second reviewer pointed out, in the discussion section, this paper should provide the reason behind the evolution of the physical and social spaces of VRCs. And in this section, we have discussed how the physical and social spaces of VRCs produced during China’s urbanization process. We found that the spatial production of VRCs is a process of “top-down” production of macrophysical space dominated by government power and multiple capital and a “bottom-up” production of microsocial space shaped by multivariate residents. In addition, given the massive influx of migrants and the changes in the occupation of landless farmers, a new socioeconomic relationship is gradually formed in a new physical space. The complex and diverse new social relationships among the residents have profound impact on social space, which is the most fundamental cause of the production of the microsocial space of VRCs. What’s more, we also found that the production of physical space of VRCs shows that the urbanization of land and population in China could be planned and manipulated by the government. However, the urbanization of individuals, which is represented by the production of social space, is difficult and challenging. The social relationship and spatial boundary of the residents in VRCs are actually typical microcosms of the urbanization of individuals lagging behind the urbanization of land and population. All these details can be found in the last section which have been marked in the revised manuscript (See lines 500-540). As the reviewer pointed out, the process of urbanization of land suitable for agriculture and the criteria for selecting urban sites are also an important aspect during the urbanization process in China. We do appreciate the reviewer’s concerning. However, for the case study of Qunyi Community, the municipal government of Kunshan has expropriated a great deal of agricultural land to construct the development zone. This process indicates that the production of physical space of VRCs is manipulated by the government. In addition, the criteria for selecting urban sites is a more macroscale topic of urban sites selection while we only focus on a relative small scale issue, i.e., the evolution of the physical and social spaces of VRC. Thus, the land suitable for agriculture and the criteria for selecting urban sites are quite beyond the scope of the current paper. Hope these clarifications can help to have a better understanding of this paper.
(6) Line 60 and 61: Review the term 'socialist'. The term 'social scientists' is probably more appropriate.
Response: Thanks for your kind reminding, the term of 'socialist' has been revised into the term of 'social scientists' in the entire revised manuscript.
(7) Figure 3 right and figure 5 need graphic scale and North.
Response: In the revised manuscript, the graphic scale bar and North compass have been added in these two figures as suggested.

Reviewer 2 Report
It is a well-written manuscript and the topic is significant in China's rapid urbanization. Before it can be accepted for publication, some problems need to be addressed:
1. The innovation/originality of this research should be highlighted. E.g., What is the difference between your work and other similar research based on the theory of production of space?
2. In section 2, theoretical framework should be separated from research context/literature review and the framework needs to be elaborated as it is the theoretical basis of this research.
3. In section 3.2, detailed information about the data collection such as how many interviewees were invited and how to select the sample should be given. In addition, a sample of the questionnaire used in the research could be attached as an appendix to make the methodology more clear.
4. In section 6, could you provide deeper implications/insights based on the evidence and findings shown in sections 4 and 5? What is the reason behind the evolution of the physical and social spaces of VRCs? More discussions should be derived from the perspective of production of space.
Author Response
It is a well-written manuscript and the topic is significant in China's rapid urbanization. Before it can be accepted for publication, some problems need to be addressed:
- The innovation/originality of this research should be highlighted. E.g., What is the difference between your work and other similar research based on the theory of production of space?
Response: We do appreciate the reviewer’s comment. From the perspective of theory of production of space, actually, the focus of our paper is not try to improve this theory or propose a new theory for the research of evolution of the physical and social spaces of VRC. However, the focus of our paper is to use this theory as basis to explore the processes and mechanisms of the physical and social space evolution of VRCs. Thus, the innovation/originality of this paper should be the physical and social space evolution of VRCs. This is because the VRC is a special urban community that the Chinese government vigorously promotes in conjunction with rapid urbanization. At present, with the increasing number and expanding scale, VRCs have become a vital component of the urban space. Compared with previous rural communities, VRCs have undergone substantial changes in the physical community form and social organization of the residents. However, VRCs are centralized resettlements of landless farmers and often present a special city–village characteristic that is very different from other urban communities. In the context of the rapid urbanization in China, VRCs have become an important platform to evaluate the adjustment relationship among the urbanization of land, population, and individuals. Thus, the innovation/originality of this paper is that we sought to study the evolution process and mechanism of the physical and social spaces of VRCs. The roles of power, capital, and social class were analyzed and their resultant impact on the community were evaluated. This research would have tremendous implications for the benign development of VRCs in China’s new type of “people-centered” urbanization. These details can be find in lines 56-94. Hope this clarification of innovation/originality of this paper can help to have a better understanding of this paper.
- In section 2, theoretical framework should be separated from research context/literature review and the framework needs to be elaborated as it is the theoretical basis of this research.
Response: Thanks for your kind suggestion. It is true that the theoretical framework should be separated from research context/literature review. Thus, in the revised manuscript, we have raised a new section of 2.3 which tries to describe the framework. In addition, the framework has been elaborated as suggested. That is, in this paper, we follow the theory of the production of space to explore the evolution of the physical and social spaces of VRCs (Figure 1). Power, capital and social class have been regarded as the key elements of production of space. Specifically, power and capital affect the land through spatial construction (i.e., planning and site selection, demolition and resettlement, and investment and construction). These two factors have been regarded as the major factors which control the production of the macrophysical space of VRCs. In addition, the diverse social classes of multiple residents affect the community through spatial reshaping (i.e., lifestyle change, rise of informal economy, and social network reconstruction). The roles of diverse social classes have been regarded as the major factors which influence the production of the microsocial space of VRCs. The roles of power, capital and class together form the conceptual framework for this study (Figure 1). With this framework, the VRC of Qunyi Community in Kunshan City is used for conducting the research on its evolution of physical and social spaces from the production of space perspective (See lines 160-172).
- In section 3.2, detailed information about the data collection such as how many interviewees were invited and how to select the sample should be given. In addition, a sample of the questionnaire used in the research could be attached as an appendix to make the methodology more clear.
Response: We fully agree with this comment from the reviewer. Actually, in the paper, we have stated that a total of 232 out of the 250 questionnaires were valid (effective rate of 92.8%), thereby meeting the sample size requirements (See lines 224-225). The interviewees were classified into four types: landless farmers, migrant workers, local New Kunshan citizens (i.e., farmers in the surrounding villages who purchased second-hand housing in Qunyi), and alien New Kunshan citizens (i.e., migrant workers who purchased second-hand housing in Qunyi). The detailed information collected through the questionnaire includes the residents’ personal attributes, personal and family members’ occupation, workplace, and income (Table 1). The housing or renting conditions are also included in the questionnaire (i.e., length of residency, housing or renting area, satisfaction with the environment, numbers of resettlement apartment, relationship with the tenant, attitude toward the tenant, and rent) (Table 1). We also surveyed the social interaction and daily activities of the residents. This detail information can be found in Lines 208-225. In addition, as the reviewer suggested, the questionnaire used in the research has been attached as an appendix in the revised manuscript. Hope these revision and clarification can help to have a better understanding of this paper.
- In section 6, could you provide deeper implications/insights based on the evidence and findings shown in sections 4 and 5? What is the reason behind the evolution of the physical and social spaces of VRCs? More discussions should be derived from the perspective of production of space.
Response: We do appreciate this comment. Actually, in the discussion section, we already provide the reason behind the evolution of the physical and social spaces of VRCs. In this section, we have discussed how the physical and social spaces of VRCs produced during China’s urbanization process. We found that the spatial production of VRCs is a process of “top-down” production of macrophysical space dominated by government power and multiple capital and a “bottom-up” production of microsocial space shaped by multivariate residents. In addition, given the massive influx of migrants and the changes in the occupation of landless farmers, a new socioeconomic relationship is gradually formed in a new physical space. The complex and diverse new social relationships among the residents have profound impact on social space, which is the most fundamental cause of the production of the microsocial space of VRCs. What’s more, we also found that the production of physical space of VRCs shows that the urbanization of land and population in China could be planned and manipulated by the government. However, the urbanization of individuals, which is represented by the production of social space, is difficult and challenging. The social relationship and spatial boundary of the residents in VRCs are actually typical microcosms of the urbanization of individuals lagging behind the urbanization of land and population. All these details can be found in the last section which have been marked in the revised manuscript (See lines 500-540). Hope these clarifications can help to have a better understanding of this paper.

Round 2
Reviewer 1 Report
The authors have partially met the requirements of this referee, providing explanations and comments on the other petitions. Therefore, all the requests are considered to have been met and rectified.